# Assembly of Microparticles to Patterned Trenches Using the Depletion Volume Effect

**DOI:** 10.3390/mi10070428

**Published:** 2019-06-28

**Authors:** Yaoki Mori, Ryota Kawai, Hiroaki Suzuki

**Affiliations:** 1Department Precision Mechanics, Faculty of Science and Engineering, Chuo University, Tokyo 112-8551, Japan; 2Precision Engineering Course, Graduate School of Science and Engineering, Chuo University, Tokyo 112-8551, Japan

**Keywords:** self-assembly, mesoscale assembly, three-dimensional assembly, depletion volume effect, excluded volume effect

## Abstract

In this paper, we demonstrate that 20 μm microbeads can be preferentially assembled into substrate trenches of similar width by employing a polymer (depletant) that induces the depletion volume effect (depletion attraction). In previous works, we proved that this strategy is useful to assemble mesoscale parts in a site-specific manner. Here, we show that it is also applicable to assemble functional parts, such as fluorescent particles, into trenches engraved on the surface of two- and three-dimensional template components. A convenient advantage of this strategy is that it is independent of material properties for assembling mesoscale functional components into desired patterns.

## 1. Introduction

The assembly of small parts or components, which are not compatible with standard photolithography-based fabrication technologies, onto a substrate to form desired patterns is a difficult task that has been receiving considerable attention in recent years. The main widespread techniques to assemble small components (i.e., microbeads) into regular patterns are microstamping [1,2] and capillary- or template-based methods [3,4,5,6,7]. They have been used in several prospective applications such as photonic crystals [8], micro lens array [9], display of structured color [10], micro/nano lithography [11,12], and bio-separation/sensing [13,14]. In the microstamping approach, components are first adhered typically on the convex part of the stamp, and then transferred by pressing onto a substrate. In the capillary-based approach, a solution of suspended components is introduced into a slit or a shallow channel, where they accumulate at the liquid–gas interface, forming a tightly packed structure. To produce an arrayed pattern, either a patterned microchannel or dimples and trenches made on the substrate, into which components are confined, is required. In addition, researchers reported the way to spatially arrange colloidal particles without lithographic processes, namely by using interparticle interactions in a nematic liquid crystal [15,16]. It has been proven that highly ordered arrays can be readily formed with these methods, even though their applications have been limited on planar substrates.

With recent advances of three-dimensional manufacturing techniques and apparatus, a development of versatile methods for assembling small components onto three-dimensional surface will be of particular interest in the fabrication of highly sophisticated devices [17,18,19,20,21]. Assembling on a 3D surface in a desired pattern may require the allocation of different material or the local modification of the surface property. Furthermore, standard photolithographic processes were developed focusing in planar surfaces, making their application difficult when curves and facets are present. Recent additive manufacturing techniques, such as stereolithography and fused deposition modeling (FDM), have provided facile means to obtain 3D components with good resolution. They, however, are limited for manufacturing employing single polymer materials. Thus, for patterning exogenous components into desired locations using affinity-based principles, pre-patterning of different materials or surface modification are still required. It is advantageous, therefore, to employ a three-dimensional (3D) self-assembly approach for assembling heterogeneous components, using a method not dependent on material’s physical properties.

In the field of colloid science, control of dispersion and aggregation of colloidal particles is important, and interactions between them and with other macro-objects have been studied extensively [22]. The van der Waals force and electrostatic interaction are major forces that depend on the physical properties of the material. In contrast, the excluded volume effect (or depletion interaction) has a different origin. This interaction becomes apparent when a large number of nanoparticles or polymer molecules are present in a medium containing larger suspended components. In this situation, a region in which those nano-objects cannot exist, within the thickness equivalent to their radius, appears around the surface of large components due to the steric repulsion (excluded volume). Since this region is as almost a pure solvent, the chemical potential is greater than the surrounding bulk polymer solution. Consequently, the contact of large components is enhanced to reduce the total excluded volume due to mutual overlapping to lower the free energy of the system [23,24].

The excluded volume effect is unique in a sense that it does only depend on the number and spatial arrangement of molecules/components present in the system. Up to now, the thermodynamic characteristics of this effect have been verified experimentally to assemble components up to several micrometers. Dinsmore and Yodh [25] investigated not only the aggregation control among them, but also the reason behind their preferential confinement at the substrate’s groove’s corner. At this region, the surface area in contact with the component increases, enhancing attraction due to the excluded volume effect when compared to a flat plane or an edge of the protruded structure. By carefully choosing nano-objects’ molecular size and concentration, therefore, tuning the extent of the interaction so that components are specifically located at the recessed part of the substrate becomes possible.

In the past literature, polymers with molecular weight of 10,000 or less and gyration radius lower than 4 nm, or nanoparticles of up to 100 nm in size, have been employed as nano-objects (depletants) for assembling components of ~1 μm [25,26,27,28]. The same strategy is not effective, however, for components larger than tens of μm [29] since the mechanism that drives collision between particles also changes. For components about 1 μm in size, the Brownian motion acts as a random driving force, carrying particles into close proximity, while for meso-scale components, it becomes less effective and forced agitation, such as tumbling or convection, is required. Previously, we showed that by using polyethylene glycol (PEG) with a molecular weight of several megadalton (gyration radius > 100 nm), microfabricated components of up to 100 μm can be coupled or allocated into specific places by shape complementarity [29,30].

In this study, using the same principle, we show that functional exogenous components with the size of several tens of μm can be specifically allocated inside grooves engraved on a template component. The components were immersed into a PEG aqueous solution, which was stirred for a long time. The extent of the bonding force can be readily controlled by adjusting the concentration of the polymer. Using this approach, we show that fluorescent particles, as a functional model component, can be assembled into trenches engraved on a sub-millimeter-sized 3D template fabricated with micro stereolithography. As this method does not depend on material properties of the assembled elements, we expect that it can be applied in the allocation of various hard and soft components such as semi-conductor, LED, cells, and gel-particles on 3D templates.

## 2. Principle and Methods

### 2.1. Principle and Design

The principle and fundamental design of the present experiment is depicted in Figure 1. Figure 1a shows a general representation of colloidal particle attraction based on the depletion volume (*V*_dep_) effect. In the environment where the depletant is present at a high concentration around spherical particles, two or more particles aggregate, creating regions of overlapping *V*_dep_. The extent of reduction of *V*_dep_ (Δ*V*_dep_) is represented in the figure by the cross hatched area. On the other hand, for the interaction between a sphere and a flat surface (Figure 1b, left), reduction of *V*_dep_ is greater when compared to the previous case. Furthermore, at the corner of a concave trench (Figure 1b, right), the overlapping volume is more than two times greater than the sphere-flat surface interaction case. Thus, by adjusting the strength of the depletion attraction and agitation, these particles’ exclusive accumulation into concave trenches becomes possible [25].

### 2.2. Fabrication and Experimental Procedure

All template components used in this study, presented in Figure 2a–c (also see Appendix A), are designed using CAD software (Rhinoceros 5.0; AppliCraft, Tokyo, Japan). The data, converted to bitmap image slices with spacing of 10 μm by employing a data preparation software (Magics, Materialise Japan, Tokyo, Japan), are processed by a 3D stereolithography tool (Acculas SI-C1000, D-MEC, Tokyo, Japan) with epoxy-based UV curable resin (KC-1257, D-MEC, Tokyo, Japan). After completion of consecutive layering and exposure steps, a glass substrate, on which stereolithography products are present, is cut into a smaller piece by a diamond cutter. Then, this piece is put inside a 50 mL plastic tube and immersed in a solvent (EE-4210, Olympus, Tokyo, Japan) for at least 30 min to remove uncured resin. After replacing the solvent with ethanol, an ultrasonic wave is applied in the bath for 10 min to release the fabricated parts from the glass substrate. Finally, ethanol is replaced by Milli-Q water to store the template components. The scanning electron microscope (SEM) images are, thereafter, obtained by VE-8800 (Keyence, Osaka, Japan).

Three different template types with trenches with designed patterns are fabricated for assembling the experiment of 20 µm fluorescent polystyrene beads (Fluorobrite YG Microspheres 20 µm with 20% C.V., Polysciences, Inc., Warrington, PA, USA). First, an appropriate PEG concentration for the present assembling system is identified by using a simple trenches’ pattern: Four straight trenches of 22 μm width engraved into a ~0.3 mm × 0.3 mm × 0.1 mm template (Figure 2a). Based on the concentration determined, assemblies of the beads onto more complex templates with dimensions of 1.38 mm × 0.6 mm or 1.52 mm× 1.12 mm, as depicted in Figure 2b, are tested. To evaluate if the procedure is possible on 3D surfaces, a 0.4 mm cubic template with different trenches’ patterns on five out of six faces is fabricated. The trench cannot be formed on the bottom face owing to the limitation of the manufacturing principle. These template components were used in the following assembling experiments after being released from the substrate.

The assembling procedure is carried out by immersing, one or several template parts together with ~10^5^ polystyrene beads (20 μm), into an aqueous solution of 4MDa PEG and molar concentrations of 0, 5, 50, or 500 nM (weight concentrations of 0, 0.02, 0.2, 2 mg/mL) in a 0.2 mL plastic PCR tube. Sodium dodecyl sulfate (SDS), an anionic amphiphile, at a concentration of 0.5% (*w*/*w*), is included in the solution to avoid hydrophobic interaction among beads and templates. This system is agitated by stirring the tube with a rotational mixer (Magic Mixer TMM-5L, Kenis, Osaka, Japan) at 30 rpm for designated time periods. The assembly of beads in trenches is, afterwards, inspected by an inverted microscopy (Nikon-Ti, Tokyo, Japan) and placing the flat face of the PCR tube down on the stage. The flow of the experimental procedure is illustrated in Appendix A.

## 3. Results

### 3.1. Dependence on the Polymer Concentration

First, the assembling experiments are conducted on four square plates, depicted in Figure 2a, with PEG at different concentrations (*c*_PEG_). Figure 3 shows typical microscope images of them after 30 min agitation. For a solution with no PEG, only a small number of beads are correctly assembled in trenches (Figure 3a). They are merely sitting in trenches by chance, so that they easily came in and out of trenches during agitation. From Figure 3b–d it is possible to observe that the number of allocated beads increase with respect to *c*_PEG_ varying from states where trenches are not fully filled (*c*_PEG_ = 5 nM, Figure 3b), almost fully filled (*c*_PEG_ = 50 nM, Figure 3c), and almost fully filled plus beads aggregates on their surroundings (*c*_PEG_ = 500 nM, Figure 3d). At higher concentration, the depletion attraction become so strong, that 20 μm beads even remains attached on flat surfaces. Above this PEG concentration, beads in the medium aggregate spontaneously [29], so that localized assembly cannot be achieve.

The average number of beads confined in the trench region per template is plotted in Figure 3e. As the trench’s length is 200 μm, only 10 beads of 20 μm diameter can be packed in one line; however, due to the slight shrinkage of resin in manufacturing, only 9 beads in one line could be packed (e.g., Figure 3c; top line). Thus, 36 beads are required to tightly and completely fill all four trenches. At *c*_PEG_ = 0 and 5 nM, the average numbers of confined beads among four plates are 13.5 and 26.5, respectively, with relatively large variation (C.V. ~ 22% and 10%, respectively). Meanwhile, at *c*_PEG_ = 50 nM, the average number of confined beads reaches ~29 (81% of the complete packing) and C.V. is reduced to ~5%, indicating reproducible results. We decided to employ *c*_PEG_ = 50 nM in all of the following experiments.

### 3.2. Assembly of Beads into the Designed Planar Trenches

Using the experimental condition determined in the previous experiment, new assembly of the same beads are conducted onto the template of more complex designed patterns. Figure 4a,b shows the confinement of beads after ~3 day of agitation, conducted in order to obtain high coverage of them. At the end of the experiment, aggregation on the flat area, especially around the letters Z, U, L, A, and B, can be seen in the bright field image (Figure 4a). We speculate that it happens because of the chance of mutual encountering of beads increases during a long agitation procedure. However, after the template is picked up by tweezers and carefully and gently shaken in the solution, aggregated beads on the flat region are readily dispersed. As a result, it is possible to recognize the shapes of all letters in the fluorescence imaging (Figure 4b), even though an unfilled area still remains.

In a similar experiment, we can confine beads into trenches forming a game character shape after ~5 days of agitation (Figure 4c). After assembly, the template is simply taken out of the PEG solution and dried in the air at room temperature. SEM observations show that monodisperse beads are allocated only in trenches (Figure 4d,e). Although no concentration’s variation tests are done for this experiment, it is believed that the agitation time, required to cover all trenches distributed over a large area, can be reduced by increasing the bead concentration.

### 3.3. Assembly of Beads into the Trenches on 3D Structure

Using an identical experimental condition to previous experiments, we conducted the assembly of the same beads onto a template with trenches in five faces of a 0.4 mm cube to prove the applicability of this approach into 3D templates. During agitation, the cube illustrated in Figure 2c tumble in a tube together with the beads. Afterward, the cube is taken out from a tube and placed in a petri dish with PEG solution for observation. Figure 5 shows the confinement of beads in trenches on five faces after 20 h of agitation. All five faces are observed under an inverted microscope by rolling with tweezers. In both bright field and fluorescent images, localized assembly of fluorescent beads in the shape of five unique faces can be clearly identified. Moreover, these beads did not come off when the cube was rotated.

In this and previous experiments, we conducted agitation for a long time (>1 day) to confirm that all trench area is filled by beads. However, we found that more than 50% of the trench area is already filled at 1.5 h in the separate experiment (Appendix A). This observation indicates as the assembly proceeds, the chance of filling the remaining space becomes smaller. In addition, we assume that increasing the concentration of beads will accelerate assembling. However, as a side effect, too many beads easily aggregate and deteriorate the assembling results as well as the visibility. These issues arise from the stochastic nature of the self-assembling system, which has to be solved in the future.

## 4. Conclusions

Here, we demonstrated the assembly of 20 μm fluorescent beads into the trenches formed on template components. In all cases (both for simple and complex 2D design patterns as well as 3D surfaces), the resulting images have shown good quality of the process. This feature’s main advantage is the surface-independent characteristic that arises from changing in the allocating mechanism from surface tension/electrostatic forces to the depletion volume effect. The method can also be applied to components made of different material properties, since any solid with non-specific interaction can be assembled by simply adding depletant in the environment. The assembly can also be controlled by altering the concentration of depletant. The present approach can be applied to weak and reversible assemblies in a material-independent manner, such as assembly of soft materials and biological cells. Although we expect that these goals could be achieved in the near future at a phenomenological level, detailed understanding of thermodynamics and depletion effect of extremely large polymers (1 MDa) is needed to further expand the possible unexpected applications [30].

## Figures and Tables

**Figure 1 micromachines-10-00428-f001:**
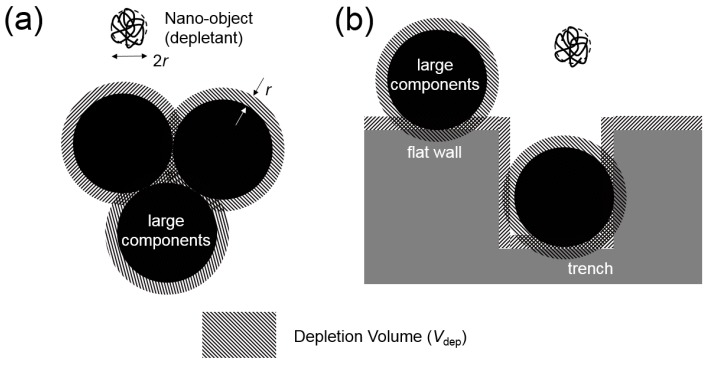
Schematic representation of the depletion volume effect. (**a**) Aggregation of spherical beads. (**b**) Attachment on flat surface (left) and confinement in the trench (right) of beads. Nano-objects (depletants) are present in abundance in the surrounding medium, but only one is depicted in the figure.

**Figure 2 micromachines-10-00428-f002:**
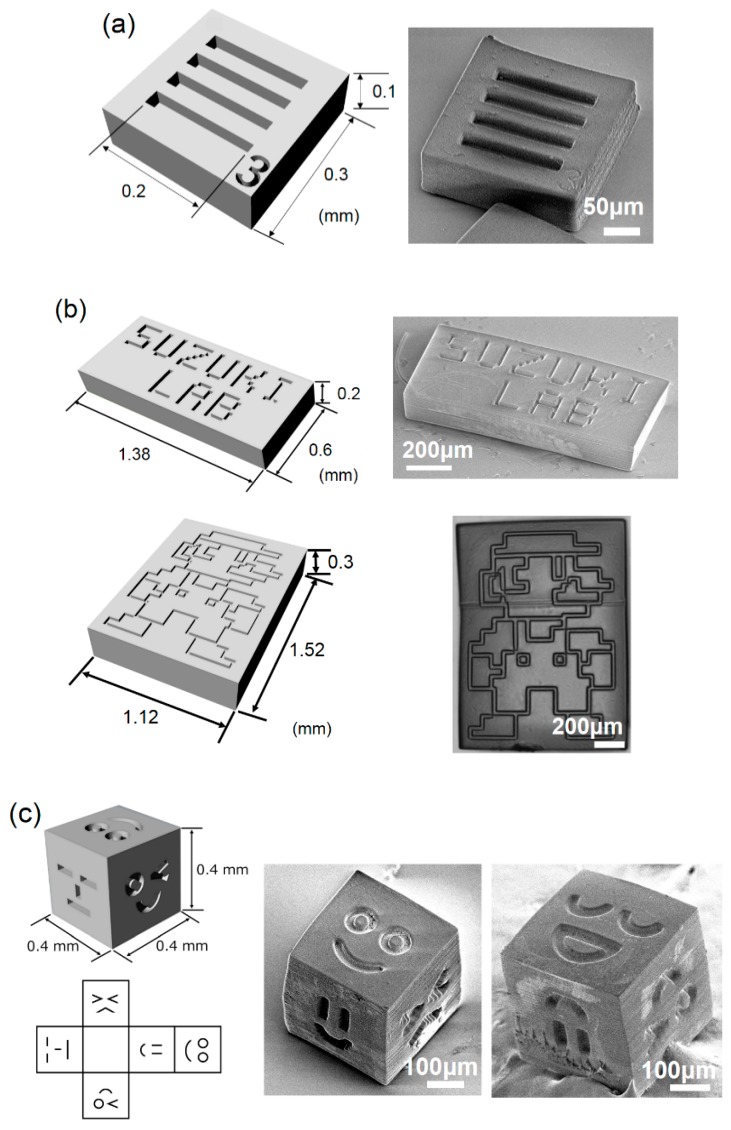
Design and real images of template components fabricated by micro stereolithography. (**a**) Square template with four straight trenches used for determining polyethylene glycol (PEG) concentration. (**b**) Rectangular flat plates with designed trenches (alphabetic letters and a game character). (**c**) A cube structure of different faces on five sides.

**Figure 3 micromachines-10-00428-f003:**
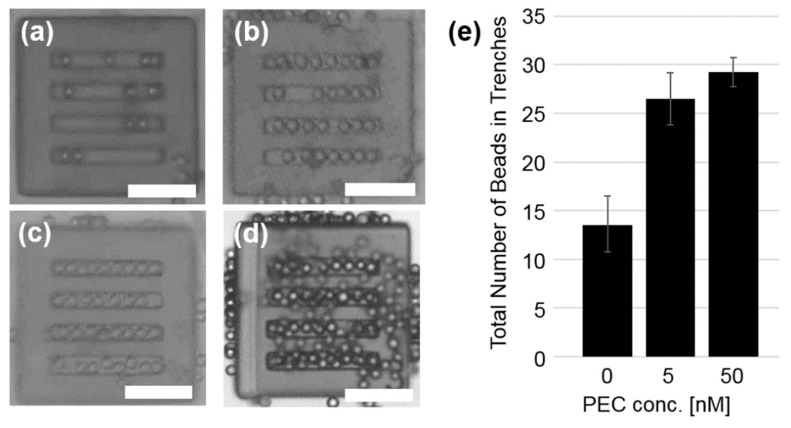
Microscope images of square templates with straight trenches agitated with polystyrene beads. Concentrations of 4MDa PEG in solutions are (**a**) 0 nM, (**b**) 5 nM, (**c**) 50 nM, and (**d**) 500 nM, respectively. Scale bars = 100 μm. (**e**) Average number of the beads confined in trenches among four template components. Data at *c*_PEG_ = 500 nM were omitted because of aggregation.

**Figure 4 micromachines-10-00428-f004:**
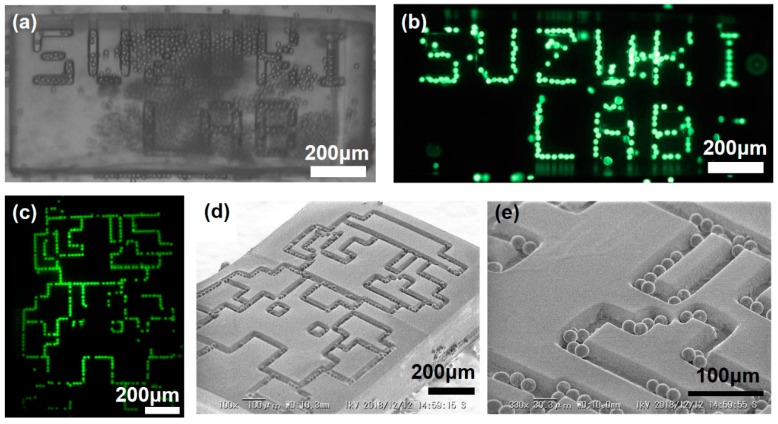
Assembly of fluorescent beads on the flat template having designed patterns trenches. (**a**) Bright field and (**b**) fluorescence microscope images of beads assembled letters. (**c**) Fluorescence microscope image and (**d**,**e**) SEM images of beads assembled to the pattern of a character.

**Figure 5 micromachines-10-00428-f005:**
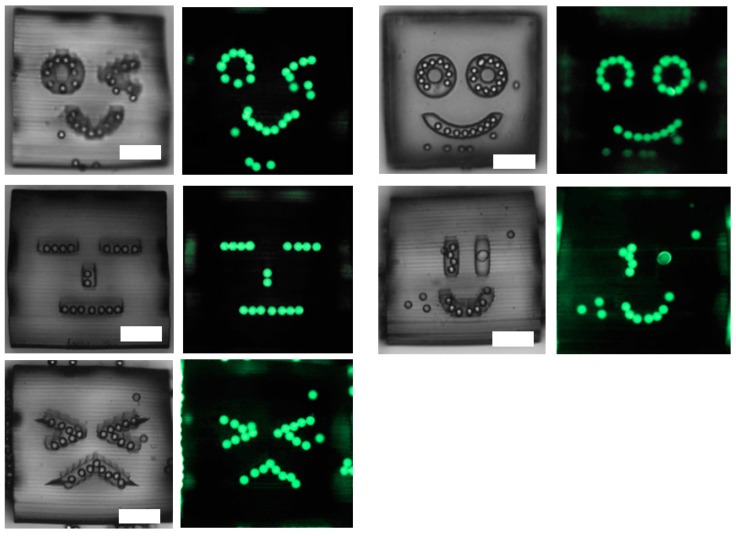
Assembly of beads on faces engraved on five faces of a cube. Left: Bright field images and Right: Fluorescent images. All scale bars represent 100 μm.

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
