# Peer review of "Assembly of Microparticles to Patterned Trenches Using the Depletion Volume Effect"

_micromachines, 2019, doi:10.3390/mi10070428_

Round 1

Reviewer 1 Report

In this manuscript, Mori et al. report on assembly of 20 µm microparticles to substrates with topographies (e.g., channels). Specifically, the authors added polymers to the continuous phase and investigate the influence of polymer concentration on the positioning of colloidal particles. The results reported in this paper seem interesting to me. However, the conclusion claimed by the authors cannot be supported by the experimental results shown in this manuscript. Therefore, I believe the following suggestions/comments should be addressed before acception/publication:

1.      The size of the fluorescent colloidal particles used in this work is 20 µm, which gives rise to significant gravitational forces that can dominate the process of colloidal assembly. The authors should provide theoretical calculation ora experimental observations (e.g., small size particles) to demonstrate that this colloidal assembly is dominated by depletion interactions.

2.      If the mechanism for colloidal assembly in this work is depletion interaction, there should be another critical concentration of polymeric depletants at which assembly of particles in bulk polymer solution will be achieved. The authors should comment on this point.

3.      Recently, people have reported that interparticle interactions in anisotropic/structural solvents, e.g., Musevic et al. Science, 2006, 313, 954, Tkalec et al. Science, 2011, 333, 62, Frederic et al. JACS, 2013, 135, 9972, Wang et al. Soft Matter, 2014, 10, 8821, Wang et al. Advanced Functional Materials, 2014, 24, 6219...... The authors should acknowledge these works in the Introduction.

4.      The authors should expand the discussion of future work in the area as the authors see it. Any other big questions for the field? Are there key issues that need to be worked out? Leaving the reader with some big picture aims, while balancing the sharing of undoubtedly excellent future work of these investigators, would make this article more stand out.

Author Response

1. The size of the fluorescent colloidal particles used in this work is 20 µm, which gives rise to significant gravitational forces that can dominate the process of colloidal assembly. The authors should provide theoretical calculation or experimental observations (e.g., small size particles) to demonstrate that this colloidal assembly is dominated by depletion interactions.

> Thank you for your note. The specific density of polystyrene beads is 1.05. We have measured the density of PEG solution and it is almost the same with that of pure water. Thus, the sedimentation velocity of 20 µm beads is estimated to be 10 µm/s in water with standard viscosity (~10-3 Pa s). We investigated the sedimentation velocity in the polymer (PEG) solution in our separate experiment (this work is just accepted to JMEMS), and it became nearly one third of that for pure water. Thus we can eliminate the possibility that only a gravitational force is responsible for assembly, because we obtained fewer assembly with no-PEG condition (Fig. 3a), in which the effect is gravity is stronger than that with PEG (Fig. 3c, d). It is difficult to directly prove that other intermolecular forces are responsible, but we rely on our previous work and others (ref. 22, 23, 25-28) that use similar experimental conditions. This main aim of this work is demonstration of the same effect to the different format, not theoretical validation.

2.      If the mechanism for colloidal assembly in this work is depletion interaction, there should be another critical concentration of polymeric depletants at which assembly of particles in bulk polymer solution will be achieved. The authors should comment on this point.

> Yes, this is exactly what we obtained in Fig. 3(d). At 500 nM, beads aggregated on the flat surfaces as well as in the bulk. The same result was observed in our previous papers (ref. 29, 30).

3.      Recently, people have reported that interparticle interactions in anisotropic/structural solvents, e.g., Musevic et al. Science, 2006, 313, 954, Tkalec et al. Science, 2011, 333, 62, Frederic et al. JACS, 2013, 135, 9972, Wang et al. Soft Matter, 2014, 10, 8821, Wang et al. Advanced Functional Materials, 2014, 24, 6219...... The authors should acknowledge these works in the Introduction.

> Thank you very much. I was not familiar with these works. I added a sentence to mention these works in Introduction.

4.      The authors should expand the discussion of future work in the area as the authors see it. Any other big questions for the field? Are there key issues that need to be worked out? Leaving the reader with some big picture aims, while balancing the sharing of undoubtedly excellent future work of these investigators, would make this article more stand out.

> Thank you very much. I believe this method will be useful for applications such as patterning of cells and other soft-materials. Actually there are remaining unclear issues regarding the thermodynamics and depletion effects of such a large polymer (> 1MDa), since most of the previous works on the depletion interaction were conducted with a PEG with moderate size (< 10 kDa). Actually we discussed the unsolved issues in our approach in our previous work which is in press in J. MEMS. (Timing of publication became the same since the review process of micromachines was extremely fast;). This paper is more on the application oriented than the basic study of mechanism and conditions. I added a sentence to mention this point and the reference to our JMEMS paper at the last part in conclusion.

Reviewer 2 Report

This paper describes how the volume depletion effect may be used to drive assembly of 20 micron scale microspheres (including microspheres with functionalities like fluorescence) selectively into certain locations on a surface.  The volume depletion effect increases with the amount of surface area that is involved in the volume depletion, and this makes assembly in certain locations more favorable than assembly in other locations.  The effect is used to enable assembly of spheres into channels that are arranged in arbitrary patterns on a surface.  The work follows on prior research in which microfabricated structures were assembled with each other rather than onto a larger host surface or substrate.

The paper is interesting and mostly well written, and it can make a useful contribution to the literature.  Before publication, a few issues should be addressed.

1.      Some of the procedures are described in an unclear manner.  For example, page 3 line 113 describes how after the layering and exposure steps are complete, a glass substrate is cut (etc.).  Please instead state the steps in order.  The layering and exposure are carried out on a glass substrate.  Then the glass substrate is cut, etc. Otherwise the reader will wonder if the glass substrate is underneath the structures created by stereolithography or if it’s something else entirely.

2.     There is another example on page 3 around line 125. It’s not clear whether the faces were formed into a cube before the assembly was carried out or whether the assembly was carried out before the faces were formed into a cube.  It eventually became clear that the cube was formed and then the assembly was carried out, but you should say this in the first place for clarity.  Also, how did you assemble the cube – manually, and with epoxy to hold it in place? 

3.     The first description of the assembly procedure (in the paragraph on page 4 that starts on line 136) is extremely unclear.  It sounds like you put the template parts, the beads, and the tube together into the PEG solution, which of course isn’t the case.  It became clear when you explained it for the second time starting in line 146 on page 5.  You should edit the paper to explain it clearly the first time, and then you won’t need to explain it a second time in the results section.  (You can still restate it if you think it helps the flow of your paper, but you won’t have to after the first explanation is clear.)

4.     Why did you use 5 nM concentration when 50 nM  concentration gave better results without any apparent down sides? 

5.     The assembly takes a very long time.  Does using a different concentration of PEG affect the time required for assembly, or is that dominated almost entirely by the circulation patterns and number of spheres rather than by the strength of the volume depletion interaction?  Please address this explicitly in the paper.

6.     When you do assembly onto the cube, does the cube tumble into different orientations, or does it retain its orientation with the same face always up?  This is important because it determines whether the assembly must have happened on vertically-oriented faces (which would be the case if the cube does not tumble) or whether it may have always occurred on whichever face was oriented horizontally at the time (which would be the case if different faces point upward at different times).  If your approach permits assembly to happen on vertically-oriented faces, that would be a valuable feature of your approach and one that would be worth highlighting, but only if it’s true.  Regardless, please describe in the paper what’s actually happening with the cube during assembly. 

Author Response

1.      Some of the procedures are described in an unclear manner.  For example, page 3 line 113 describes how after the layering and exposure steps are complete, a glass substrate is cut (etc.).  Please instead state the steps in order.  The layering and exposure are carried out on a glass substrate.  Then the glass substrate is cut, etc. Otherwise the reader will wonder if the glass substrate is underneath the structures created by stereolithography or if it’s something else entirely.

> Thank you for pointing out. I modified this part in the revised manuscript. I added the illustration of the experimental procedure in the Supplementary Figure S2.

2.     There is another example on page 3 around line 125. It’s not clear whether the faces were formed into a cube before the assembly was carried out or whether the assembly was carried out before the faces were formed into a cube.  It eventually became clear that the cube was formed and then the assembly was carried out, but you should say this in the first place for clarity.  Also, how did you assemble the cube – manually, and with epoxy to hold it in place?

> Cubes with faces (trenches) were directly formed by stereolithography, and they were not fixed. Cubes were agitated freely in solution together with beads. I did my best to clarify these points in the revised manuscript. Epoxy is a material composing the template, not an adhesive.

3.     The first description of the assembly procedure (in the paragraph on page 4 that starts on line 136) is extremely unclear.  It sounds like you put the template parts, the beads, and the tube together into the PEG solution, which of course isn’t the case.  It became clear when you explained it for the second time starting in line 146 on page 5.  You should edit the paper to explain it clearly the first time, and then you won’t need to explain it a second time in the results section.  (You can still restate it if you think it helps the flow of your paper, but you won’t have to after the first explanation is clear.)

> Thank you. This part was revised by the editing company (Editage). I also thought it became odd, but I assumed it could be readable for English speakers. I changed the text and simplified the second explanation.

4.     Why did you use 5 nM concentration when 50 nM concentration gave better results without any apparent down sides?

> I am very sorry about that. It was 50 nM. Somehow zero was omitted during the writing by mistake.

5.     The assembly takes a very long time.  Does using a different concentration of PEG affect the time required for assembly, or is that dominated almost entirely by the circulation patterns and number of spheres rather than by the strength of the volume depletion interaction?  Please address this explicitly in the paper.

> We believe that increasing the concentration of beads makes assembling faster. In the present condition, the number density of beads is not high (5 x 105 /ml; the volume fraction of 0.2%), which is fairly dilute so that structure was visible through a light microscope. Too many beads easily aggregate and deteriorate the assembling results as well as the visibility. I mentioned this tradeoff in the revised manuscript.

6.     When you do assembly onto the cube, does the cube tumble into different orientations, or does it retain its orientation with the same face always up?  This is important because it determines whether the assembly must have happened on vertically-oriented faces (which would be the case if the cube does not tumble) or whether it may have always occurred on whichever face was oriented horizontally at the time (which would be the case if different faces point upward at different times).  If your approach permits assembly to happen on vertically-oriented faces, that would be a valuable feature of your approach and one that would be worth highlighting, but only if it’s true.  Regardless, please describe in the paper what’s actually happening with the cube during assembly.

> It does tumble in the cube together with beads. I believe that assembly is taking place in every direction, but cannot directly prove it because we only observe the cube after sedimenting to the bottom. I added a sentence to describe this fact.

Round 2

Reviewer 1 Report

I have carefully read the revised manuscript and the letter of response. The reviewers have addressed all my questions and I believe the manuscript is now ready for publication in Micromachines.

Author Response

Thank you very much for reviewing.

Reviewer 2 Report

The revisions have mostly addressed the issues brought up in review, and the writing is now appropriately clear.  There is one remaining issue.  The original comment copied in below was asking about how the concentration of the _PEG_ affected speed of assembly, not how the concentration of microspheres affected speed of assembly.  This question should still be addressed.

Quoting from original review:  “5.     The assembly takes a very long time.  Does using a different concentration of PEG affect the time required for assembly, or is that dominated almost entirely by the circulation patterns and number of spheres rather than by the strength of the volume depletion interaction?  Please address this explicitly in the paper.”

Author Response

I see. Actually this point is difficult to answer with simple yes or no. When PEG conc is high enough, beads aggregate just by themselves in the medium. It occurs immediately, so we will not get neat alignment only in trenches. I will see if I can insert this information into the current flow of the manuscript. Thak you very much for the valuable comment.